# A Latent Generative Model for Closed-set and Open-set Recognition

## Abstract

The classic recognition problem assumes that all possible classes in testing are known in advance during training, which can be termed closed-set recognition (CSR). As a natural extension, open-set recognition (OSR) requires models to reject samples of unknown classes that are not encountered in the training phase. Traditional discriminative models struggle to learn decision boundaries for OSR due to the absence of unknown samples. This has led to existing methods focusing on either CSR or OSR, as optimizing one often results in performance degradation of the other. In this paper, we offer a formalization for OSR based on learning theory, demonstrating that CSR and OSR share the same goal for generative models. Motivated by this core insight, we introduce a neural Latent Gaussian Mixture Model (L-GMM) accompanied by a collaborative training algorithm. The model consists of an encoder that maps inputs to a latent space, and a density estimator that computes probability densities. The end-to-end training algorithm, designed in a collaborative manner, learns the density estimator through maximum likelihood estimation and trains the encoder using a discriminative loss derived from the generative model. This framework yields a model capable of performing both CSR and OSR. Experimental results show that L-GMM outperforms its discriminative counterparts in image recognition and segmentation in CSR with models trained from scratch. These models also outperform other specialized methods when directly applied to OSR without any modifications or prior knowledge.

## 1 Introduction

In recognition problems, most models operate under the closed-set assumption [44; 28; 80], *i.e.*, all test samples are drawn from known classes that have been seen in the training phase. In open-set scenarios, however, test samples from unknown classes should be rejected [10; 72; 91; 4; 76; 103; 11]. The fundamental challenge of OSR lies in the unobservability of unknown data distribution.

Since it is infeasible to learn feature representations of unknown data, typical OSR solutions train discriminative models with cross-entropy loss on known classes. During testing, a thresholding of the softmax probability is employed to decide if a sample should be rejected. Although some variants have been developed to better utilize the softmax scores [24; 65], these methods generally face two limitations: (1) learning decision boundaries between known classes may not be sufficient to identify outlier classes [103; 6; 104; 37], and (2) for these methods to function, samples of unknown classes should exhibit a uniform probability distribution over the known classes [11]. Consequently, most existing methods consider CSR and OSR as tradeoffs [65; 110] and approach them separately.

In this paper, we reexamine the nature of OSR and its relation to CSR. We formulate the risk of both problems (§3) in a more principled manner than existing formalizations [76; 11]. By investigating the risks, we show that **the goals of CSR and OSR are identical for generative models**: both risks can be minimized by applying maximum likelihood estimation on training data of known classes (§ 3.3). Discriminative probabilities for recognition can thereby be obtained via the Bayes rule.

As it is practically challenging to learn an almost perfect generative model in high-dimensional spaces, we propose a latent generative model with a collaborative training algorithm for both CSR and OSR for real-world data (§2). The model is composed of two parts: an encoder that maps the input sample to a latent space, and a density estimator that outputs a probability of the latent variable. This model offers two advantages. (1) The latent variable may lie in a lower-dimensional

space compared to the input, hence its distribution might be approximated more precisely. (2) We can assume a closed-form distribution for the latent variable by adding constraints to the training process so that we can easily compute the probability. Specifically, we use a neural Gaussian Mixture Model as the density estimator for the latent variable, naming our method L-GMM.

This latent generative model is deeply integrated with a collaborative training scheme. The fully end-to-end framework is driven by two forces. In a single pass, the **generative learning** part learns the density estimator by maximum likelihood estimation, and the **discriminative learning** part trains the encoder. In this way, the two components are decoupled but highly synchronized by the latent representation, which needs to attain both a generative capacity and a discriminative power. The discriminative power establishes the basis for CSR, and the generative capacity makes OSR possible. This framework can also maximize (1) the divergence between the latent densities of different classes, and (2) the mutual information between latent variables and output classes.

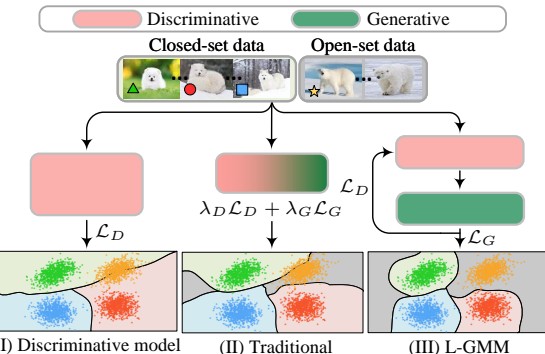

Figure 1: **Discriminative**: most recognition models focus on modeling the decision boundaries between known classes and struggles with OSR problems. **Traditional**: generative models for recognition are usually trained with a mixed objective. **L-GMM**: our latent generative model (§2.2) preserves the generative nature that models the data density and handles both CSR and OSR. The yellow scatter points stand for open-set data, while the other three colors (*i.e.*, green, blue, and red) represent closed-set data.

Our experiments show that L-GMM is effective on both CSR and OSR. In §5.1, with ResNet [29] and Swin [58] backbone architectures, L-GMM outperforms its discriminative counterparts on closed-set image classification, by *training from scratch*. Using the same model instance trained previously, in §5.2 we show competitive results on open-set image recognition tasks. We present similar results on closed-set and open-set image semantic segmentation in §5.3 and §5.4.

Overall, this paper makes three main contributions. **First**, we formulate the learning-theoretic risks for both CSR and OSR problems and show that generative models minimize both risks by MLE on known classes, advocating learning a single model for both scenarios. **Second**, we design a latent generative framework that integrates a latent generative model with a collaborative training scheme. **Third**, we demonstrate advanced performance on both CSR and OSR tasks by one single model instance of L-GMM, a concrete example of the proposed framework. Our code will be released.

## 2 METHODOLOGY

We aim to build a recognition model that can be directly used in open-set scenarios after training with closed-set datasets. Intuitively, generative models may be preferable over discriminative ones because they learn the boundaries of distributions. In this section, we propose our latent generative model and its accompanied training scheme. In §3, we will show that generative models learned by maximum likelihood estimation minimize the recognition risks for both CSR and OSR.

### 2.1 LATENT GENERATIVE MODELS WITH COLLABORATIVE TRAINING

We can obtain a discriminative probability from a generative model via the Bayes rule:

$$p(y|x) = \frac{p(x|y)p(y)}{\sum_y p(x|y)p(y)}, \tag{1}$$

where $x$ is a data sample and $y$ is a class label. Since the prior probabilities $p(y)$ are typically set as uniform distributions (also in our case), the core part is to learn the data distribution $p(x|y)$. However, modeling the distribution of real-world data can be challenging due to its high dimensionality and complexity. To alleviate this, we propose a latent generative model composed of two parts: (1) an encoder $f_\phi(\cdot)$ that maps the input $x$ to a latent variable $z = f_\phi(x)$, and (2) a probabilistic generative model $p_\theta(z|y)$ that outputs a probability density of the latent variable given the label $y$.

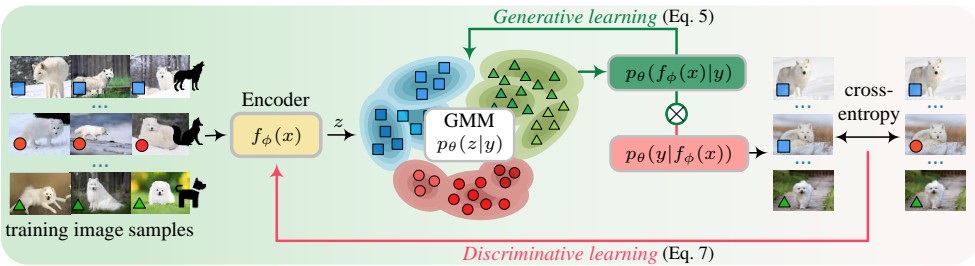

Figure 2: In a single pass of L-GMM, the density estimator is updated by generative learning, and the encoder is driven by discriminative learning. Thus the framework learns a latent representation with the benefits of both generative and discriminative models through collaborative training.

The above model structure has three advantages. (1) We can constrain the latent variable to conform to any desired distributions. (2) $z$ may have a lower dimensionality compared to $x$, hence the distribution might be approximated more precisely by the density estimator. (3) We can assume a closed-form distribution for $z$ so that we can easily compute the probability as well as add constraints to the training process. Ideally, the encoder produces discriminative features for both CSR and OSR, and the density estimator lays the foundation to recognize samples from unknown classes.

We propose to learn this latent generative model through a collaborative training scheme by adding an auxiliary discriminative loss. In a single pass, the model parameters $\theta$ and $\phi$ are updated concurrently, driven by two forces to exploit the strengths of both generative and discriminative models:

**Generative Learning**. We update $\theta$ of the density model $p_\theta(z|y)$ by:
$$\text{maximize}_\theta \, \mathrm{E}_{p(x,y)}[\log p_\theta(z = f_\phi(x)|y)], \tag{2}$$
where $p(x, y)$ denotes the true distribution of $(x, y)$, which can be approximated by sample average.

**Discriminative Learning**. We update $\phi$ of the encoder $f_\phi(x)$ by:
$$\text{maximize}_\phi \, \mathrm{E}_{p(x,y)}[\log p_\theta(y|z = f_\phi(x))], \tag{3}$$
which is equivalent to minimizing the cross-entropy loss on the discriminative probability.

This algorithm learns an intermediate representation that achieves a generative capacity as well as a discriminative power. Intuitively, the generative objective empowers the model with open-set robustness, and the discriminative objective learns a distinctive latent representation. This framework also maximizes (1) the divergence between the latent densities of different classes, and (2) the mutual information between latent variables and classes. Please see the supplementary for more details.

**Discussion.** When considering the hybrid training of the generative and discriminative model, existing methods [97; 84; 103; 25] typically update the entire model using a loss function of the form $\lambda_D \mathcal{L}_D + \lambda_G \mathcal{L}_G$, where the subscripts $D$ and $G$ stand for discriminative and generative components with weight factors $\lambda_D$ and $\lambda_G$ (Figure 1). In contrast, we adopt a different approach by updating the generative model solely through the unified objective (MLE) with the underlying discriminative features learned by, while keeping the entire framework end-to-end. This novel combination preserves the generative nature of our architecture with additional discriminative power.

## 2.2 LATENT GAUSSIAN MIXTURE MODEL (L-GMM)

As a concrete realization, we use the Gaussian Mixture Model (GMM) as the generative model $p_\theta(z|y)$ of the latent space and obtain L-GMM (Figure 2). The reason is threefold: (1) GMM is a universal approximator for densities, (2) GMM has a closed-form formulation and guarantees $\int_\mathcal{X} p(x|y)dx = 1$ hence minimizes the risk ( Proposition 1), and (3) the multi-modal nature of GMM avoids mode collapse of the latent space. Specifically, L-GMM computes a closed-form density:
$$p_\theta(z|y) = \sum_k p(k|y; \pi_y)p(z|k; \boldsymbol{\mu}_{yk}, \boldsymbol{\sigma}_{yk})$$
$$= \sum_k \pi_{yk}\mathcal{N}(z; \boldsymbol{\mu}_{yk}, \boldsymbol{\sigma}_{yk}), \tag{4}$$
where $k|y \sim Mult(\pi_k)$ is the prior probability of assigning $z$ to mixture component $k$, i.e., $\sum_k \pi_{yk} = 1$. $\boldsymbol{\mu}_{yk}$ and $\boldsymbol{\sigma}_{yk}$ are the mean and covariance matrix for the $k$-th component of class $y$. With the prior distribution $p_\theta(y) = \pi_y$, we have the parameters of the generative model defined:

$\theta = \{\pi_y, \pi_{yk}, \boldsymbol{\mu}_{yk}, \boldsymbol{\sigma}_{yk}\}_{y,k}$. Here $\pi$ can be estimated by maximum likelihood, which can be computed by counting the frequency of examples that fall into different classes/mixture components, and $\boldsymbol{\mu}, \boldsymbol{\sigma}$ can be updated by gradient-based algorithms. We design the collaborative training algorithm for L-GMM as follows to learn a non-degenerate GMM with discriminative power.

**Generative Learning.** We update $\theta$ of the generative model by minimizing:

$$\mathcal{L}_G = -\mathcal{L}_{mle} + \lambda_{one}\mathcal{L}_{one} + \lambda_{avg}\mathcal{L}_{avg}, \tag{5}$$

where $-\mathcal{L}_{mle}$ is the negative log-likelihood, $\lambda_{one}$ and $\lambda_{avg}$ are the coefficients for the regularizing loss functions. The regularizer $\mathcal{L}_{one}$ computes the mean squared error between the best component assignment (*i.e.*, a one-hot vector) and the actual assignment, and $\mathcal{L}_{avg}$ computes the Wasserstein distance $\mathcal{W}(\cdot)$ between $\pi$ and the uniform distribution:

$$\begin{aligned}
\mathcal{L}_{mle} &= \sum\nolimits_{(x,y)} \log\left(\sum\nolimits_k \pi_{yk}\mathcal{N}(f_\phi(x); \mu_{yk}, \sigma_{yk})\right), \\
\mathcal{L}_{one} &= \sum\nolimits_{(x,y,k)} (p(k|f_\phi(x), y) - \mathbb{1}(k = k^*))^2, \\
\mathcal{L}_{avg} &= \sum\nolimits_y \mathcal{W}(p(k|y), U(K)).
\end{aligned} \tag{6}$$

Here $\mathbb{1}$ is the indicator function, and $k^*$ is the mixture component that has the highest probability. $U(K)$ is a discrete uniform distribution of $K$ mixture components. $\mathcal{L}_{one}$ encourages a sample to be assigned to only one component, and $\mathcal{L}_{avg}$ encourages the data samples to be evenly distributed among mixture components. Intuitively, $\mathcal{L}_{one}$ and $\mathcal{L}_{avg}$ are adding constraints to the Gaussian mixture model to avoid the model collapsing into a single Gaussian distribution, so that it can better capture the underlying modes of the latent representation from the same class.

**Discriminative Learning.** We update $\phi$ of the encoder by minimizing the cross-entropy loss:

$$\mathcal{L}_D = -\sum_{(x,y)} \log \frac{\sum\limits_{k=1}^{K} \pi_{yk}\mathcal{N}(f_\phi(x); \boldsymbol{\mu}_{yk}, \boldsymbol{\sigma}_{yk})}{\sum\limits_{y'=1}^{C} \pi_{y'} \sum\limits_{k=1}^{K} \pi_{y'k}\mathcal{N}(f_\phi(x); \boldsymbol{\mu}_{y'k}, \boldsymbol{\sigma}_{y'k})}. \tag{7}$$

During inference, the out-of-detection data are recognized by applying a threshold to the probability.

L-GMM with the above training scheme can be applied to both CSR and OSR. (1) By learning a generative density on the latent representation, L-GMM is built with the capability to recognize samples of unknown classes. (2) The feature space is discriminatively trained end-to-end under the guidance of the generative classifier, hence L-GMM learns a powerful representation for recognition.

## 3 CLOSED-SET RECOGNITION AND OPEN-SET RECOGNITION

In this section, we revisit CSR and OSR from a learning-theoretic perspective, which provides the core insight that motivates our method in §2: generative models minimize the risk for both tasks.

### 3.1 EXISTING PROBLEM FORMULATION AND ITS LIMITATIONS

The OSR problem was initially formulated in [76]. Let $f \in \mathcal{H}$ be a model in a function space $\mathcal{H}$ and $f_y(x)$ is the confidence of an input $x$ being class $y$. The authors define an open space risk as

$$R_\mathcal{O}(f_y) = \frac{\int_\mathcal{O} f_y(x)dx}{\int_{S_\mathcal{O}} f_y(x)dx}, \tag{8}$$

where $\mathcal{O}$ is the open space that is sufficiently far from any known positive samples, and $S_\mathcal{O}$ is a ball that includes all of the positive examples as well as the open space. The open space risk is a relative measure of positively labeled open space compared to the overall measure of positively labeled space. Then the goal of OSR is to find a recognition model that minimizes an open set risk:

$$\arg\min_{f \in \mathcal{H}} R_\mathcal{O}(f_y) + \lambda_r R_\mathcal{E}(f_y), \tag{9}$$

where $R_\mathcal{E}(f_y) = \frac{1}{n}\sum_{i=1}^{n} L(f(x_i), y_i)$ is the empirical risk on the closed-set data, $L(\cdot)$ is a loss function, and $\lambda_r$ is a regularization coefficient. However, this risk definition has several limitations. (1) The ratio form of $R_\mathcal{O}$ is inconsistent with $R_\mathcal{E}$. (2) $R_\mathcal{O}$ is not principled since it ignores the loss function by giving a fixed form of the risk. (3) Most importantly, it defines the CSR objective $R_\mathcal{E}$ as a tradeoff; it is a regularization term in OSR, which may be a biased view of both CSR and OSR.

## 3.2 A Learning-theoretic Formulation

We formulate a risk definition extending the traditional one in learning theory [89] from CSR to OSR, providing a different and more principled perspective on this problem. In CSR, a model $f(\cdot)$ is trained to predict whether an observation $x \in \mathcal{X}_\mathcal{C}$ belongs to a class $y \in \mathcal{C}$. Here $\mathcal{C}$ is the set of known discrete classes seen in the training phase and $\mathcal{X}_\mathcal{C}$ is the corresponding input space. In many cases, the model $f$ approximates a probability distribution, which can be either a discriminative $p(y|x)$ or a generative model $p(x|y)$. In learning theory, the risk is defined as the expected loss:

$$R_{csr}(f) = \mathrm{E}_{p(x,y)}[L(f_y(x), y)] = \int_\mathcal{C} \int_{\mathcal{X}_\mathcal{C}} p(x,y) L(f_y(x), y) dx dy, \tag{10}$$

where some common choices for the loss function $L(\cdot)$ are the cross-entropy loss for discriminative models and the negative log-likelihood for generative models. To find the best $f$ that minimizes the risk, the true risk is approximated in training by the empirical risk $R_\mathcal{E}$. It can be shown that the empirical risk converges to the true risk when we have enough training data [89], under the assumption of CSR that all test inputs come from the same distribution as the training samples.

In OSR, test inputs may come from unseen classes $\mathcal{U}$, making $x \in \mathcal{X}_\mathcal{C} \cup \mathcal{X}_\mathcal{U}$ and $y \in \mathcal{C} \cup \mathcal{U}$. Now:

$$
\begin{aligned}
R_{osr}(f) &= \int_{\mathcal{C} \cup \mathcal{U}} \int_{\mathcal{X}_\mathcal{C} \cup \mathcal{X}_\mathcal{U}} p(x,y) L(f_y(x), y) dx dy \\
&= \underbrace{\int_\mathcal{C} \int_{\mathcal{X}_\mathcal{C}} p(x,y) L(f_y(x), y) dx dy}_{R_{csr}} + \underbrace{\int_\mathcal{U} \int_{\mathcal{X}_\mathcal{U}} p(x,y) L(f_y(x), y) dx dy}_{R_{gap}}
\end{aligned}
\tag{11}
$$

where the first term $R_{csr}$ can be approximated by the empirical risk, and the second term $R_{gap}$ is the gap between CSR and OSR. Therefore models trained with closed-set datasets that minimize $R_{emp}$ are not guaranteed to minimize $R_{osr}$.

One important question here is: what is a good loss function that involves unknown classes? Since the goal of OSR is to reject samples from unknown classes, the loss function should output a high cost when the model assigns a high probability for the sample to be any known class. We define:

**Definition 1.** A loss function $L$ is *open-set safe* if it satisfies the following condition. Consider any sample $x \in \mathcal{X}_\mathcal{U}$ of an unknown class $y \in \mathcal{U}$, we have: $f^* = \arg\min_{f \in \mathcal{H}} L(f(x), y) \iff \forall y_c \in \mathcal{C}, f^* = \arg\min_{f \in \mathcal{H}} f_{y_c}(x)$.

For example, an open-set safe $L$ for discriminative models $f_y(x) = q(y|x)$ can be:

$$L(f(x), y) = \begin{cases} -\log q(y|x) & x \in \mathcal{X}_\mathcal{C}, y \in \mathcal{C}, \\ \max_{y_c \in \mathcal{C}} \log q(y_c|x) & x \in \mathcal{X}_\mathcal{U}, y \in \mathcal{U}. \end{cases} \tag{12}$$

One can switch the max operator with sum or average operators, the loss functions would also be open-set safe: any sample of unknown classes still should have a low score for any known class.

## 3.3 Generative Models Minimize Recognition Risk

In this section, we show that the goals of CSR and OSR are aligned for generative models.

**Proposition 1.** *For generative models $f_y(x) = q(x|y)$ that approximates the true distribution $p(x|y)$, the OSR risk $R_{osr}$ with an open-set safe loss can be minimized asymptotically by MLE on data of known classes:* $\arg\max_{f \in \mathcal{H}} \int_\mathcal{C} \int_{\mathcal{X}_\mathcal{C}} p(x,y) \log f_y(x) dx dy = \arg\min_{f \in \mathcal{H}} R_{osr}(f).$

*Proof.* The MLE objective directly minimizes $R_{csr}$ by minimizing $R_{emp}$. Since $R_{osr} = R_{csr} + R_{gap}$, the proposition holds if MLE also minimizes $R_{gap}$. For the training data $x \in \mathcal{X}_\mathcal{C}, y \in \mathcal{C}$, we have

$$f^* = \arg\max_f \mathrm{E}_\mathrm{P}[\log f_y(x)] = \arg\max_f \mathcal{H}(p(x|y)) - D_{KL}(p\|q) = \arg\min_f D_{KL}(p\|q) \tag{13}$$

where $\mathcal{H}$ denotes the information entropy and $\mathcal{H}(p(x|y))$ is a constant. We can see that $f^*$ also minimizes $D_{KL}(p\|q)$. Therefore $f^* = q^*(x|y) = p(x|y)$ on known classes for $x \in \mathcal{X}_\mathcal{C}$ and $y \in \mathcal{C}$.

Suppose we test $f^*$ in the OSR setting, *i.e.*, $x \in \mathcal{X}_\mathcal{C} \cup \mathcal{X}_\mathcal{U}$. Consider a sample $x \in \mathcal{X}_\mathcal{U}$ and $y \in \mathcal{C}$:

$$q^*(x|y) \leq \int_{\mathcal{X}_\mathcal{U}} q^*(x|y) dx = 1 - \int_{\mathcal{X}_\mathcal{C}} q^*(x|y) dx = 1 - \int_{\mathcal{X}_\mathcal{C}} p(x|y) dx = 0. \tag{14}$$

For $f^*$, we have $\forall x \in \mathcal{X}_\mathcal{U}, y \in \mathcal{C}, q(x|y) = 0$. Thus $f^*$ minimizes the open-set safe loss $L$ according to Definition 1 for $y \in \mathcal{U}$, and thereby minimizes $R_{gap}$ and $R_{osr}$ (Equation 11). $\square$

## 4 RELATED WORK

**Open-set Recognition**. Existing solutions to OSR can be classified into two categories: discriminative and generative methods. **Discriminative methods**, prior to the advent of deep learning, exhibited subpar performance without meticulous feature engineering [77; 78; 102; 39]. Recent deep-learning-based models bring more appealing results, and can be categorized into two groups. **I. Classification-based methods** largely rely on classifiers. Methods such as [31] firstly proposed the detection of open-set examples by demonstrating that anomalous samples have a lower maximum softmax probability than in-distribution samples and [53] introduced ODIN to enable more-effective detection from gradient information. Other methods refer to outlier exposure to help models learn open/closed-set discrepancy [32; 12], which requires a re-training step on classification, resulting in performance degradation. **II. Distance-based methods** uses different distance metrics, *i.e.*, radial basis function kernel [88], Euclidean distance [35] or KL distance [33] to identify out-of-distribution examples. **Generative methods**, on the other hand, falls into **density-based methods**, which explicitly model the in-distribution with some probabilistic models, and leave test data in low-density regions. Methods such as flow-based methods [42; 40] estimate the in-distribution directly, identify out-of-distribution examples by likelihoods, and classify examples using discriminative models.

With a collaborative training strategy, L-GMM gains benefit from its generative nature and therefore handles open-set problems naturally, with neither external datasets of outliers [111; 32], nor specifically designed distance metrics [85; 15]. It also differs from most uncertainty classification-based methods that utilize post-processing to adjust the prediction scores of softmax-based classification networks[32; 26; 71]. The most relevant ones to our work are density-based models[103; 70], which measure the likelihood ratio of samples directly w.r.t. data distribution. However, they are built upon pre-trained representation [7] or specialized for OSR [103], ignoring the closed-set performance.

**Generative *vs* Discriminative Classifiers**. Generative and discriminative classifiers represent two ways of solving classification tasks [61]. Generally, the generative classifiers (*e.g.*, naive Bayes) learn the class densities $p(x|y)$, while the discriminative classifiers (*e.g.*, softmax) learn the class boundaries $p(y|x)$ without regard to the underlying class densities. In practical classification tasks, softmax discriminative classifiers are used extensively [61], due to their simplicity and excellent performance. Nonetheless, generative classifiers have several advantages over their discriminative counterparts [9; 52] (*e.g.*, accurately modeling the input distribution, and explicitly identifying unlikely inputs in a natural way). Some of the recent work [3; 14] therefore investigates the potential (and the limitation) of generative classifiers in adversarial example defense[81; 51; 23], explainable AI[61], out-of-distribution detection [79; 63; 38; 5], and semi-supervised learning[64; 36].

The discriminative models and the generative models are mutually related [45; 62]. According to [45], the only difference between these models is their statistical parameter constraints. Intuitively, given a generative model, we can derive a corresponding discriminative model, which makes it possible to get the best of two worlds by training both models jointly. The hybrid training procedure has long been claimed even before the deep learning evaluation [45; 69]. However, hybrid training approaches continue to encounter several limitations that prevent their widespread implementation in both closed and open-set scenarios: Methods like [27] are discriminatively trained; some provide limited performance on naive datasets [64; 69]; [90; 56] are kernel-based methods which simply applied the last layer of DNN as the GMM representation; [74; 20] separately train DNNs for feature representations, which are then fed into independently trained GMM. More importantly, most of these methods focus on single (open/closed-set) task [64; 86; 67], ignoring the availability proofs on both OSR and CSR tasks. The experiences from previous arts serve as the impetus for our current work, which proposes accommodating both OSR and CSR simultaneously.

## 5 EXPERIMENTS

We respectively examine the performance and robustness of L-GMM on CSR (§5.1, §5.3) and OSR (§5.2, §5.4) on image classification and segmentation. For both tasks, we first train the model from scratch under the CSR setting. Then we directly apply the trained model to OSR problems without further changes or fine-tuning. Overall, the experiments demonstrate that L-GMM performs better than its discriminative counterparts on CSR and other competitive methods on OSR.

### 5.1 CLOSED-SET IMAGE CLASSIFICATION

**Datasets.** The evaluation for closed-set image classification is carried out on three commonly used datasets, *i.e.*, CIFAR-10 [43], CIFAR-100 [43] and ImageNet [73].

Table 1: **Closed-set image classification top-1 accuracy** on CIFAR-10 [43] `test` and CIFAR-100 [43] `test` (§5.1). The error bars are based on three randomized runs (same for Table 2)).

| Dataset | Method | Backbone | top-1 |
|---------|--------|----------|-------|
| CIFAR-10 | ResNet [29] | ResNet50 | 95.55 ± (0.09)% |
| | **L-GMM**-ResNet | | **95.67** ± (0.08)% |
| | ResNet [29] | ResNet101 | 95.58 ± (0.10)% |
| | **L-GMM**-ResNet | | **95.77** ± (0.09)% |
| CIFAR-100 | ResNet [29] | ResNet50 | 79.81 ± (0.12)% |
| | **L-GMM**-ResNet | | **79.98** ± (0.08)% |
| | ResNet [29] | ResNet101 | 79.83 ± (0.11)% |
| | **L-GMM**-ResNet | | **80.15** ± (0.10)% |

Table 2: **Closed-set image classification top-1 and top-5 accuracy** on ImageNet [73] `val` (see §5.1). Further results on alternative encoders are available in Appendix.

| Method | Backbone | top-1 | top-5 |
|--------|----------|-------|-------|
| ResNet [29] | ResNet101 | 77.52 ± (0.12)% | 93.06% ± (0.10)% |
| **L-GMM**-ResNet | | **77.83** ± (0.12)% | **93.20**% ± (0.09)% |
| Swin [58] | Swin-B | 83.36 ± (0.10)% | 96.44% ± (0.08)% |
| **L-GMM**-Swin | | **83.47** ± (0.09)% | **96.71**% ± (0.08)% |

**Network Architecture.** L-GMM is crafted on CNN-based ResNet50/101 [29] and Transformer-based Swin-Small/Base [58]. We remove the last linear classification layer and add a simple convolutional layer to reduce the dimension to 128. This acts as the feature encoder that maps the input samples to a latent space (§2.2). For the density estimator, we directly optimize the GMM parameters (§2.2) by backpropagation. The *default* configurations are adopted for training from scratch.

**Results.** Table 1 compares L-GMM with its discriminate counterpart on CIFAR-10 and CIFAR-100 `test`, based on the most representative CNN network architecture, *i.e.*, ResNet. As seen, L-GMM gains consistently better performance than its discriminative counterpart: L-GMM is **0.12**% higher on ResNet50, and **0.19**% higher on ResNet101. Similarly on CIFAR-100, L-GMM is **0.17**% higher on ResNet50, and **0.32**% higher on ResNet101. We further show comparison results on ImageNet `val` on Table 2. L-GMM shows strong performance over various network architectures. Specifically, in terms of `top-1` acc., our L-GMM surpasses the discriminative counterpart by **0.32**% on ResNet101. L-GMM also gives compelling performance over Transformer architecture, *i.e.*, **83.47**% *vs* 83.36% on Swin-B. We provide corresponding error bars by training three times, with different initialization seeds in Table 1 and Table 2. With the same backbone architecture and training settings, one can safely attribute the closed-set performance gain to L-GMM.

## 5.2 OPEN-SET IMAGE RECOGNITION

We evaluate the performance of our L-GMM on standard datasets used for open-set recognition and compare with state-of-the art methods. The results include performance on out-of-distribution recognition and open-set recognition tasks, respectively.

**Datasets.** Following common practices, we evaluate on five out-of-distribution datasets (*i.e.*, TinyImageNet (Crop) [47], TinyImageNet (Resize) [47], LSUN (Crop) [100], LSUN (Resize) [100] and iSUN [96]), and two open-set recognition datasets (*i.e.*, CIFAR+10 [43] and TinyImageNet [47]).

**Experiment Protocol.** For out-of-distribution recognition, we use the models trained in the closed-set setting (§5.1,Table 1): ResNet101 trained on CIFAR-10 and CIFAR-100 `train` only, respectively. For open-set recognition, all results are applied to ResNet34 as the encoder backbone following common practices [25; 83; 11].

**Evaluation Metrics.** In Table 3, we apply the area under receiver operating characteristics (AUROC), and false positive rate (FPR95) at a true positive rate of 95%. In supplementary §C.3, we provide results on out-of-distribution recognition using additional evaluation metrics. We follow [31; 49] for the experimental setup. The AUROC is also applied in open-set recognition tasks.

**Results.** On Table 3, we show an overall comparison of various methods that are trained with/without out-of-distribution data with five out-of-distribution benchmark datasets. In particular, we consider maximum softmax probability (MSP) [31], ODIN* [34], KL Matching [33] and ODIN [53]. For fairness, methods other than ODIN do not incorporate out-of-distribution data for tuning. Following [34], ODIN* is the modified version that does not need any out-of-distribution data for tuning while ODIN refers to out-of-distribution data during inference. The results show that L-GMM provides competitive performance on out-of-distribution detection, it reaches first or second place

Table 3: **Out-of-distribution recognition results** for in-distribution datasets CIFAR-10 [43] and CIFAR-100 [43] on five out-of-distribution datasets. ODIN* is the modified version of ODIN provided in [34] that does not need any out-of-distribution data for tuning. All values are percentages averaged over three runs, and the best results are indicated in **bold**. Additional results on out-of-distribution data using other evaluation metrics are available in supplementary (see §5.2).

| ID | OOD | AUROC ↑ | FPR95 ↓ |
|---|---|---|---|
| | | **Methods**: MSP / ODIN* / KL Matching / ODIN / **Ours** | |
| CIFAR-10 | iSUN [96] | 93.99 / 93.70 / 89.72 / 94.49 / **95.62** | 45.51 / 37.01 / 52.69 / 31.60 / **28.96** |
| | LSUN (C.) [100] | 93.73 / 94.05 / 90.16 / 93.77 / **94.45** | 43.31 / 33.88 / 46.62 / 34.82 / **30.30** |
| | LSUN (R.) [100] | 89.96 / 90.88 / 86.89 / 92.24 / **92.83** | 43.43 / 37.16 / 46.96 / 31.82 / **26.60** |
| | TinyImg. (C.) [47] | 93.30 / 93.49 / 90.67 / **94.11** / 93.53 | 47.46 / 39.57 / 53.79 / 35.42 / **34.38** |
| | TinyImg. (R.) [47] | 92.91 / 92.66 / 89.03 / 93.48 / **93.54** | 50.31 / 43.98 / 54.13 / 38.14 / **34.61** |
| CIFAR-100 | iSUN [96] | 79.29 / 81.80 / 77.31 / 83.70 / **85.24** | 76.78 / 76.51 / 75.29 / 71.43 / **69.07** |
| | LSUN (C.) [100] | 75.49 / **83.21** / 76.31 / 82.64 / 81.88 | 80.69 / 74.06 / 79.47 / 75.04 / **73.53** |
| | LSUN (R.) [100] | 73.29 / 82.24 / 78.32 / 84.11 / **85.90** | 82.90 / 75.57 / 78.35 / 70.51 / **67.36** |
| | TinyImg. (C.) [47] | 74.71 / 84.24 / 71.01 / 85.51 / **87.84** | 81.34 / 68.64 / 81.03 / 64.54 / **60.58** |
| | TinyImg. (R.) [47] | 80.34 / 82.94 / 78.21 / 84.84 / **87.17** | 78.81 / 72.65 / 77.31 / 67.16 / **62.96** |

Table 4: **Open-set classification results** on CIFAR+10 [43] and TinyImageNet [47].

| Dataset | MSP [31] | OpenMax [5] | G-OpenMax [24] | OSRCI [65] | CROSR [98] | CGDL [84] | GFROR [67] | **Ours** |
|---|---|---|---|---|---|---|---|---|
| CIFAR+10 [43] | 0.677 | 0.695 | 0.675 | 0.699 | - | 0.681 | 0.831 | **0.833** |
| TinyImageNet [47] | 0.577 | 0.576 | 0.580 | 0.586 | 0.589 | 0.653 | 0.657 | **0.681** |

among all methods introduced in Table 3. More impressively, it even outperforms ODIN [53], which is a gradient-based method that refers to out-of-distribution data for calibration during inference.

For completeness on the open-set problems, we further compare our L-GMM with [5; 24; 65; 98; 84; 67] on two standard OSR datasets: CIFAR+10 and TinyImageNet [47] in Table 4. Following common practices [103; 67; 25], we report AUROC scores on the detection of known and unknown samples. The results show that L-GMM does enjoy strong performance gain with other state-of-the-art methods, while benefiting from an elegant, single model for scenarios.

## 5.3 CLOSED-SET IMAGE SEGMENTATION

**Datasets.** The evaluation for semantic image segmentation is carried out on two datasets: ADE20K [109] and Cityscapes [19].
**Architecture.** L-GMM is evaluated on the renowned segmentation models: DeepLab$_{V3+}$ [13] and Segformer [95], using ResNet101 [29] and MiT [95] as backbones, respectively. For fairness, all models are trained by standardized hyper-parameters [55; 93; 92].
**Results.** Table 5 demonstrates our quantitative results. We include five widely recognized methods [59; 105; 107; 82; 16] for a complete experiment setup. Our L-GMM outperforms its discriminative counterparts across two datasets, *i.e.*, with DeepLab$_{V3+}$:

Table 5: **Closed-set semantic segmentation results** on ADE20K [109] `val` and Cityscapes [19] `val` with `mIOU`. *: pre-trained on ImageNet 21K; ⋆: utilizing a larger crop size, *i.e.*, $640 \times 640$.

| Method | Backbone | ADE20K | Citys. |
|---|---|---|---|
| FCN [59] | ResNet101 | 39.9% | 75.5% |
| PSPNet [105] | ResNet101 | 44.4% | 79.8% |
| SETR [107] | ViT$_{Large}$* | 48.2% | 79.2% |
| Segmenter [82] | ViT$_{Large}$* | 51.8%⋆ | 79.1% |
| MaskFormer [16] | Swin$_{Base}$* | 52.7%⋆ | - |
| DeepLab$_{V3+}$ [13] | ResNet101 | 45.5% | 80.6% |
| **L-GMM**-DeepLab$_{V3+}$ | | **46.4%** | **81.3%** |
| Segformer [95] | MiT$_{Base}$ | 50.0% | 82.0% |
| **L-GMM**-Segformer | | **50.7%** | **82.5%** |

46.4% *vs* 45.5% on ADE20K and 81.0% *vs* 80.6% on Cityscapes, and other competitive methods. Similar performance, *i.e.*, **50.7%** *vs* 50.0% and **82.5%** *vs* 82.0% on two datasets are also obtained with Segformer architecture.

## 5.4 OPEN-SET IMAGE SEGMENTATION

**Datasets.** We apply Fishyscapes Lost&Found [7] and Road Anomaly [54] for evaluation.
**Experiment Protocol.** Following [31; 41; 57], we adopt ResNet101-DeepLab$_{V3+}$ architecture. For completeness, we also report the results on MiT$_{Base}$-Segformer. All models are initially trained in §5.3 and do not require further change for open-set image segmentation.
**Evaluation Metrics.** Following the standard practice [41; 94; 52; 7], we use three evaluation metrics in Table 6: the area under receiver operating characteristics (AUROC), the average precision (AP), and the false positive rate (FPR95) at a true positive rate of 95%.
**Results.** As shown in Table 6, based on DeepLab$_{V3+}$ architecture, L-GMM provides advanced

Table 6: **Open-set segmentation results** on Fishyscapes Lost&Found [7] and Road Anomaly [54]. ⋆: methods with confidence derived from a generative formulation (see §5.4).

| Method | DeepLab$_{V3+}$ | Extra Resyn. | OOD Data | mIOU | Fishyscapes Lost&Found | | | Road Anomaly | | |
|---|---|---|---|---|---|---|---|---|---|---|
| | | | | | AUROC↑ | AP↑ | FPR95↓ | AUROC↑ | AP↑ | FPR95↓ |
| SynthCP [94] | ✔ | ✔ | ✔ | 80.6 | 88.34 | 6.54 | 45.95 | 76.08 | 24.86 | 64.69 |
| SynBoost [22] | ✔ | ✔ | ✔ | - | 96.21 | 60.58 | 31.02 | 81.91 | 38.21 | 64.75 |
| MSP [31] | ✔ | ✗ | ✗ | 80.6 | 86.99 | 6.02 | 45.63 | 73.76 | 20.59 | 68.44 |
| Entropy [31] | ✔ | ✗ | ✗ | 80.6 | 88.32 | 13.91 | 44.85 | 75.12 | 22.38 | 68.15 |
| SML [41] | ✔ | ✗ | ✗ | 80.6 | 96.88 | 36.55 | 14.53 | 81.96 | 25.82 | 49.74 |
| Mahalanobis⋆ [49] | ✔ | ✗ | ✗ | 80.6 | 92.51 | 27.83 | 30.17 | 76.73 | 22.85 | 59.20 |
| **L-GMM**-DeepLab$_{V3+}$⋆ | ✔ | ✗ | ✗ | 81.3 | 97.31 | 45.42 | 14.15 | 85.01 | 34.73 | 48.21 |
| **L-GMM**-Segformer⋆ | ✗ | ✗ | ✗ | 82.5 | 97.76 | 48.75 | 13.21 | 89.41 | 58.13 | 45.29 |

Table 7: **Diagnostic experiments** for L-GMM (see §5.5).

| L-GMM | top-1 |
|---|---|
| Generative-only | 78.08% |
| Collab. training | **80.15%** |

(a) L-GMM training

| L-GMM | top-1 | Collapse |
|---|---|---|
| ResNet101 + GMM | 77.73% | ✔ |
| Collab. training | **80.15%** | ✗ |

(b) Collaborative training

| Loss Components | | | top-1 |
|---|---|---|---|
| $\mathcal{L}_{mle}$ | $\mathcal{L}_{one}$ | $\mathcal{L}_{avg}$ | |
| ✔ | | | 79.97% |
| ✔ | ✔ | | 80.02% |
| ✔ | ✔ | ✔ | **80.15%** |

(c) Loss components

| # Components $G$ | top-1 |
|---|---|
| $G = 1$ | 79.99% |
| $G = 2$ | 80.04% |
| $G = 3$ | **80.15%** |
| $G = 4$ | 80.09% |

(d) Gaussian components

results over all the competitors under the same setting, *i.e.*, neither external out-of-distribution data nor an additional resynthesis module is applied. [31; 41; 48] are methods based on pre-trained discriminative segmentation models, requiring post-calibration during open-set segmentation. L-GMM, on the other hand, derives confidence scores directly from likelihood. Mahalanobis [48] similarity is a method that also models data density. However, it constructs over pre-trained feature space with a single Gaussian component per class, ignoring the inner distribution of each class [106; 27; 52; 2]. When adopting SegFormer, better performance is achieved.

## 5.5 DIAGNOSTIC EXPERIMENTS

We ablate core designs of L-GMM, using ResNet101 [29] on CIFAR-100 [43]. We follow the standard training settings introduced in §5.1.

**Generative-only L-GMM *vs* L-GMM.** We first investigate the necessity of utilizing collaborative training in Table 7(a). By adding the discriminative learning component, we observe a clear performance improvement from the method with only generative learning (*i.e.*, top-1 acc.: 78.08% → 80.15%). This proves that the collaborative training scheme is practically useful for learning a distribution with high divergences between the classes, and hence improving the recognition results.

**Collaborative Training.** We further investigate the effectiveness of the training strategy. We study a variant where a neural GMM is directly fitted onto the feature space pretrained by a softmax classifier, *i.e.*, the original RestNet101. In Table 7(b), We observe a clear performance drop, *i.e.*, top-1 acc.: 80.15% → 77.73%. This indicates that a better latent representation is learned by collaborative training. We also find that the examples are concentrated on a single Gaussian component per class, indicating the necessity of generative learning to distribute examples evenly since discriminative models ignore within-class variation and collapse into a single component.

**Loss Components**. In Table 7(c), we further study the impact on three losses: $\mathcal{L}_{mle}$, $\mathcal{L}_{one}$ and $\mathcal{L}_{avg}$ introduced in §2.2. A clear performance gain is observed (*i.e.*, 79.97% → 80.15%) with the aid of $\mathcal{L}_{avg}$ and $\mathcal{L}_{mle}$, which are incorporated to better capture the modes of data during training.

**Number of Gaussian Components.** We study the impact on the number of Gaussian components in Table 7(d). When $G = 1$, each class is following the concept of unimodality, without considering within-class variation. When increasing $G$ from 1 to 3 leads to better performance (*i.e.*, 79.99% → 80.15%). This supports our hypothesis that one single Gaussian component is insufficient to either capture the underlying data distribution or consider within-class variation. We stop using $G > 3$ since the performance reduces owing to overparameterization.

## 6 CONCLUSION

In this work, we present a generic solution for CSR and OSR by means of L-GMM, which consists of a latent generative model empowered by collaborative training. Our method has two advantages: (1) the probability density model learns the data distribution that enables OSR, and (2) the feature encoder learns the discriminative power and thus achieves promising CSR results. Exhaustive experiments on two computer vision tasks validate the competitive performance of L-GMM in both CSR and OSR settings with the single model instance trained in the closed-set setting.

## REPRODUCIBILITY STATEMENT

To help readers reproduce our results, we have described the implementation details and provided pseudo-code in §G. We will release our source code after acceptance. All the datasets we use are publicly available.

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

- §A extends our discussion on the proposed latent generative framework.

- §B shows more information and results on closed-set image classification.

- §C shows more information and results on open-set image recognition.

- §D provides more information on closed-set image segmentation.

- §E provides more information and qualitative results on open-set image segmentation.

- §G presents the pseudo code and reproducibility of our code.

- §H is the discussion of legal/ethical considerations and limitations.

## A  MORE DISCUSSION ON THE FRAMEWORK

### A.1  LATENT GENERATIVE MODEL VIA COLLABORATIVE TRAINING

Now we seek to understand the collaborative training algorithm. The generative learning part is equivalent to minimizing the Kullback-Leibler divergence:

$$\text{minimize}_\theta \, D_{\text{KL}}(p(z|y)\|p_\theta(z|y)), \tag{15}$$

where $z = f_\phi(x)$, and $p(y|z)$ is the true distribution.

To understand the discriminative learning part. In our case, $z$ is low-dimensional. Assume $p(z|y)$ is closedly approximated by $p_\theta(z|y)$ then this part minimizes the conditional entropy over $\phi$:

$$\text{E}_{p(x,y)}[\log p_\theta(y|z = f_\phi(x))] \approx -\mathcal{H}(p(y|z = f_\phi(x))). \tag{16}$$

That is, we want to choose $\phi$ so that the conditional data distribution of $y$ given $z = f_\phi(x)$ has the lowest entropy or uncertainty.

For simplicity, we first assume $p(y) = 1/C$ is uniform over all the $C$ categories. We will discuss the more general case later. For notational simplicity, let $p(y)p(z|y)$ be the data distribution of $(y, z = f_\phi(x))$. Then minimizing the conditional entropy of $p(y|z)$ is minimizing:

$$
\begin{aligned}
&\text{E}_{p(y,z)}[-\log p(y|z)]\\
&= \text{E}_{p(y,z)}\left[-\log \frac{p(y)p(z|y)}{p(z)}\right]\\
&= \log C + \frac{1}{C}\sum_y \text{E}_{p(z|y)}\left[-\log \frac{p(z|y)}{p(z)}\right]\\
&= \log C - \frac{1}{C}\sum_y D_{\text{KL}}(p(z|y)\|p(z)),
\end{aligned}
\tag{17}
$$

where $p(z) = \sum_y p(z|y)p(y)$ is the mixture of the $C$ class densities. Thus minimizing the conditional entropy amounts to maximizing $\sum_y D_{\text{KL}}(p(z|y)\|p(z))/C$, where $p(z) = \sum_y p(y)p(z|y) = \frac{1}{C}\sum_y p(z|y)$. This is a generalized version of Jensen-Shannon divergence (JS divergence) $JSD(P\|Q) = \frac{1}{2}(D(P\|M) + D(Q\|M))$, where $M = \frac{1}{2}(P + Q)$. That is, we want to find $z = f_\phi(x)$ so that the divergence between the class densities $p(z|y)$ is maximized.

In an open-set setting, suppose there are $C_{\text{total}}$ categories, and the $C$ categories in the training set is a random sample from the $C_{\text{total}}$ categories. Then the JS divergence calculated for the $C$ categories can be considered an approximation or estimation of the divergence calculated for all the $C_{\text{total}}$ categories.

In the above derivation, we assume a uniform prior distribution over classes $p(y) = 1/C$. For a more general prior class distribution, we have

$$
\begin{aligned}
& \mathrm{E}_{p(y,z)}[-\log p(y|z)] \\
&= \mathrm{E}_{p(y,z)}\left[-\log \frac{p(y)p(z|y)}{p(z)}\right] \\
&= \mathrm{E}_{p(y)}[-\log p(y)] + \mathrm{E}_{p(y)}\mathrm{E}_{p(z|y)}\left[-\log \frac{p(z|y)}{p(z)}\right] \\
&= \mathcal{H}(p(y)) - \mathrm{E}_{p(y)}D_{\mathrm{KL}}(p(z|y)\|p(z)).
\end{aligned}
\tag{18}
$$

The above is a more general version of the JS divergence.

We can also show that the discriminative learning part maximizes the mutual information between $y$ and $z$, since

$$
\begin{aligned}
& \mathrm{E}_{p(y,z)}[-\log p(y|z)] \\
&= \mathrm{E}_{p(y,z)}\left[-\log \frac{p(y,z)}{p(z)}\right] \\
&= \mathrm{E}_{p(y,z)}\left[-\log \frac{p(y,z)}{p(z)p(y)} - \log p(y)\right] \\
&= \mathcal{H}(p(y)) - D_{\mathrm{KL}}(p(y,z)\|p(y)p(z)) \\
&= \mathcal{H}(p(y)) - I(y,z),
\end{aligned}
\tag{19}
$$

where $I(y,z)$ is the mutual information.

## A.2 More on L-GMM

The discussed qualities greatly distinguishes L-GMM from most existing GMM-based neural classifiers, which are either ignoring the joint optimization of DNN features together with the GMM backend [30; 99; 74; 20] or building a GMM in the feature space on a pre-trained discriminative classifier [49; 108; 50].

## A.3 More on Generative Classifiers

Early works such as [66; 87] compared the properties of generative classifiers *vs* discriminative classifiers in theory and through experiments, with the agreement on the advantages of generative classifiers. Works like [8; 6] presented models that combine the aspects of generative and discriminative classifiers, to reach a more favorable trade-off compared to each extreme. However, these works do not consider complex tasks, and with the unmatched performance later delivered by deep-learning-based discriminative classifiers in the 2010s, generative classifiers became rarely used. Till recently, some of the deep learning literature [3; 14] studies the potential (and limitations) of generative classifiers in various fields discussed in §4.

## B Closed-set Image Classification

### B.1 Datasets

We show additional information on the closed-set classification datasets we applied in L-GMM.

- **CIFAR-10** [43] contains 60K (50K/10K for `train`/`test`) $32\times32$ colored images of 10 classes.
- **CIFAR-100** [43] contains 60K (50K/10K for `train`/`test`) $32\times32$ colored images of 100 classes.
- **ImageNet** [73] contains 1.2M images for `train` and 50K images for `validation` of 1K classes.

### B.2 Detailed Training Procedures

We use `mmclassification`[1] as the codebase and adopt the *default* training settings. For CIFAR-10, we train ResNet for 200 epochs, with batch size 16. For ImageNet, we train 100 and 300 epochs

---

[1]https://github.com/open-mmlab/mmclassification

with batch size 16 for ResNet and Swin, respectively. The initial learning rates of ResNet and Swin are set as 0.1 and 0.0005, scheduled by a step policy and polynomial annealing policy, respectively. The memory size for L-GMM models is set as 2000 examples per class [93; 52]. All other hyper-parameters are empirically set by default. All models are trained *from scratch* on eight Tesla V100 GPUs.

### B.3 ADDITIONAL RESULTS AND DIAGNOSTIC STUDY

Table 8 reports closed-set classification performance on ImageNet [73], using ResNet50 [29] and Swin-S [58] architectures. As can be seen, L-GMM again attributes decent performance. In particular, our L-GMM is **0.31**% and **0.15**% higher on ResNet50 and Swin-S, respectively.

We further study the influence of output dimensionality discussed in §6.1 from our paper. We follow our diagnostic study using ResNet101 [29] on CIFAR-100 [43] for consistency. The number of Gaussian components $G$ is set to $G = 3$ and we remain other experimental settings the same. In Table 9, with the dimension reduced to 128, it is enough for L-GMM to model the data distribution precisely, a higher dimension (*i.e.*, dimension=256) reaches the performance saturating point.

Table 8: **Closed-set image classification top-1 and top-5 accuracy** on ImageNet [73] `val` with standard deviation error bars on three runs with different initialization seeds.

| Method | Backbone | top-1 | top-5 |
|---|---|---|---|
| ResNet [29] | ResNet50 | $76.20 \pm (0.10)$% | 93.01% |
| L-GMM-ResNet | | **76.51** $\pm (0.09)$% | **93.03**% |
| Swin [58] | Swin-S | $83.02 \pm (0.14)$% | 96.29% |
| L-GMM-Swin | | **83.17** $\pm (0.14)$% | **96.42**% |

## C  OPEN-SET IMAGE RECOGNITION

### C.1  EVALUATION METRICS

Here we present the evaluation metrics applied in Table 1 from our paper, and Table 10.

- **True negative rate (TNR) at 95% true positive rate (TPR).** Let $TP$, $TN$, $FP$, and $FN$ denote true positive, true negative, false positive and false negative, respectively. We measure $TNR = TN/(FP + TN)$, when $TPR = TP/(TP + FN)$ at 95%.

- **Area under the receiver operating characteristic curve (AUROC).** It describes the relation between $TPR$ and $FPR$ interpreted as the probability of a positive sample being assigned a higher score than a negative sample.

- **Area under the precision-recall curve (AUPR).** The PR curve is a graph plotting the precision = $TP/(TP + FP)$ against recall = $TP/(TP + FN)$ by varying a threshold. AUPR-In (or AUPR-Out) is AUPR where in- (or out-of-) distribution samples are specified as positive.

- **Detection error.** It measures the probability of misclassifying a sample when the $TPR$ is at 95%. Assuming that a sample has equal probability of being positive or negative in the test, it is defined as $0.5(1 - TPR) + 0.5FPR$

### C.2  DATASETS

We provide additional information on the open-set setting, including both out-of-distribution datasets and open-set datasets. Each out-of-distribution input is pre-processed by default settings [32; 57; 34; 33; 53]: subtracting the mean of in-distribution data and dividing the standard deviation. All the datasets considered are listed below:

- **iSUN.** iSUN [96] dataset is a subset of SUN images. The entire collection of 8925 images in iSUN are included and resized to size $32 \times 32$.

- **LSUN (Crop) and LSUN (Resize).** Large-scale Scene UNderstanding (LSUN) dataset has 10000 images test set of 10 different scenes [100]. LSUN (Crop) and LSUN (Resize) are two datasets constructed by either randomly cropping image patches of size $32 \times 32$ or downsampling images to size $32 \times 32$.

Table 9: **Output dimensionality** of L-GMM

| Output dimensionality $d$ | top-1 |
|---|---|
| $d = 64$ | 79.91% |
| $d = 128$ | **80.15**% |
| $d = 256$ | 80.10% |
| $d = 512$ | 80.04% |

Table 10: **Open-set recognition results** for in-distribution datasets CIFAR10 [43] and CIFAR-100 [43] on five out-of-distribution datasets with evaluation metrics AUPR In, AUPR Out and Detection Error. ↑ indicates larger value is better, and ↓ indicates lower value is better. All values are percentages averaged over three runs, and the best results are indicated in **bold**.

| ID | OOD | AUPR In↑ | AUPR Out↑ | Detection Error↓ |
|---|---|---|---|---|
| | | **Methods**: MSP / ODIN* / KL Matching / ODIN / **Ours** | | |
| | iSUN [96] | 95.63/95.10/87.67/95.63/**95.64** | 90.21/91.70/87.87/93.00/**93.71** | 11.41/12.31/14.31/11.59/**9.92** |
| | LSUN (C.) [100] | 95.02/**94.75**/91.54/94.41/93.92 | 91.53/92.76/89.31/92.53/**93.96** | 11.74/12.35/12.89/12.75/**10.83** |
| CIFAR-10 | LSUN (R.) [100] | 95.46/94.37/91.10/94.96/**95.91** | 91.55/92.29/89.43/87.89/**94.82** | 11.14/12.45/13.01/11.70/**9.30** |
| | TinyImg. (C.) [47] | **94.86**/94.31/91.35/94.78/92.60 | 90.69/91.94/88.03/**92.88**/92.83 | 11.95/12.80/13.21/12.30/**11.38** |
| | TinyImg. (R.) [47] | **94.56**/93.59/90.56/94.25/92.60 | 89.96/90.88/86.89/92.24/**92.83** | 12.25/13.69/13.34/12.93/**11.42** |
| | iSUN [96] | 81.26/85.53/80.01/86.90/**88.84** | 75.54/75.11/71.39/78.24/**80.03** | 26.87/24.66/27.12/23.18/**22.40** |
| | LSUN (C.) [100] | 77.89/**85.63**/75.24/84.96/84.06 | 71.66/**79.06**/69.67/78.49/78.48 | 30.09/**23.46**/28.37/23.80/25.48 |
| CIFAR-100 | LSUN (R.) [100] | 77.66/84.83/80.26/86.32/**88.49** | 66.75/77.81/73.89/80.65/**82.63** | 31.84/24.48/31.58/23.03/**21.85** |
| | TinyImg. (C.) [47] | 77.09/86.21/76.31/87.26/**89.86** | 70.92/81.11/68.21/82.92/**85.20** | 31.16/23.38/31.10/22.31/**20.12** |
| | TinyImg. (R.) [47] | 82.66/85.22/80.34/86.75/**89.30** | 75.28/79.14/74.61/81.96/**84.38** | 25.23/24.21/25.49/22.74/**20.74** |

- **TinyImageNet (Crop) and TinyImageNet (Resize).** TinyImageNet dataset is a subset of ImageNet [44] which consists of 10000 test images from 200 different classes. Similar to LSUN, two datasets, TinyImageNet (Crop) and TinyImageNet (Resize) are constructed by randomly cropping or downsampling the LSUN testing set to $32 \times 32$, respectively.

For open-set datasets, we include CIFAR+10 [43] and TinyImageNet [47]. Specifically, CIFAR+10 uses data from both CIFAR10 and CIFAR100. 4 classes are sampled from CIFAR10 and unknown classes are randomly selected from CIFAR100 dataset. TinyImageNet is a subset of ImageNet consisting of 200 classes. 20 classes are randomly sampled as known and the remaining classes are set as unknown.

## C.3 ADDITIONAL RESULTS

We present additional results on other evaluation metrics introduced in §C.1. Table 10 considers CIFAR-10 [43] and CIFAR-100 [43] as the in-distribution dataset and evaluates on AUPR In, AUPR Out and Detection Error metrics. Our method handles out-of-distribution detection naturally without any modifications on either reaching the external datasets of outliers, or having additional image resynthesis structures, showed competitive results against other out-of-distribution methods discussed in §5.2 from our paper.

OpenHybrid [103] acts as the most relevant one to our work as a density-based model, which measures the likelihood ratio of samples directly w.r.t. data distribution. We therefore further compare L-GMM to [103] for out-of-distribution detection. Note that [103] lacks a code release, we then follow and design extra experiments in Table 11 to test the performance of L-GMM on CIFAR-10 [43] and CIFAR-100 [43] without any post-processing, respectively. We directly adapt the trained L-GMM model (*i.e.*, L-GMM ResNet101 [29] trained on CIFAR-10 [43] and CIFAR-100 [43], respectively) in §6.1 and follow the same experimental setup in our paper (§6.2) for open-set image recognition. We report AUROC for consistency to [103]. A stronger performance to [103] can be observed with a single model instance.

## D  CLOSED-SET IMAGE SEGMENTATION

### D.1 DATASETS

Two widely applied semantic segmentation datasets are conducted in our experiments.

Table 11: **AUROC on OpenHybrid [103] and ours** between CIFAR-10 [43] and CIFAR-100 [43]. All values are in percentage.

| Train/Test(Out-of-distribution) | OpenHybrid [103] | L-GMMM (Ours) |
|---|---|---|
| CIFAR-10 [43]/CIFAR-100 [43] | 95.1 | **95.7** |
| CIFAR-100 [43]/CIFAR-10 [43] | 85.6 | **86.4** |

Figure 3: Qualitative results (§E.2) of open-set segmentation heatmaps on Fishyscapes Lost&Found [7] `val`.

- **ADE20K** [109] has 20K/2K/3K general scene images for `train/val/test` of 150 semantic categories.
- **Cityscapes** [19] has 2,975/500/1,524 urban scene images for `train/val/test` of 19 classes.

## D.2 DETAILED TRAINING PROCEDURES

We adopt `mmsegmentation`[2] as the codebase, and follow the default training settings. We train DeepLab$_{V3+}$[13] with ResNet101 using SGD optimizer with an initial learning rate 0.1, and Segformer [95] with MiT$_{Base}$ using AdamW with an initial learning rate 6e-5. The learning rate is scheduled following a polynomial annealing policy. As common practices [95; 16], we train the model on ADE20K `train` with crop size $512 \times 512$ and batch size 16; on Cityscapes `train` with crop size $769 \times 769$ and batch size 8. The model is trained for 160K iterations on ADE20K and 80K iterations on Cityscapes. Standard data augmentation techniques, such as scale and color jittering, flipping, and cropping are used.

## E OPEN-SET IMAGE SEGMENTATION

### E.1 DATASETS

Two popular open-set segmentation datasets are conducted in our experiments.

- **Fishyscapes Lost&Found** [7] is built upon the original Lost&Found [68] dataset, which has 100/275 `val/test` images. The dataset is collected with the same setup as Cityscapes [19].
- **Road Anomaly** [54] contains 60 images where there exist anomalous objects (*e.g.*, animals, rocks, and etc.) in unusual road conditions with a resolution of $1280 \times 720$.

### E.2 QUALITATIVE RESULTS

In Figure 3, we visualize the score heatmaps generated by MSP [31]-DeepLab$_{V3}$ [13] and L-GMM-DeepLab$_{V3}$, respectively. The softmax based counterpart becomes overconfident on predictions, failing to recognize out-of-distribution examples. L-GMM, on the other hand, naturally rejects them (red colored regions).

## F RUNTIME ANALYSIS

The inference speed of L-GMM on ResNet101 ImageNet [73] `val` is 211 fps, which yields negligible overhead w.r.t the discriminative counterpart, *i.e.*, 217s $vs$ 238 fps. On DeepLab$_{V3}$ ADE20K [109] `val`, the inference speed of L-GMM is 13.21 fps, slightly slower than its discriminative counterpart, *i.e.*, 14.37 fps $vs$ 15.56 fps.

---

[2]https://github.com/open-mmlab/mmsegmentation

## G  PSEUDO CODE OF L-GMM AND REPRODUCIBILITY

The pseudo-code of L-GMM is given in Algorithm 1. L-GMM is implemented in Pytorch. Training and testing are conducted on eight Tesla NVIDIA V100 GPUs. We will release our code publicly to guarantee our reproducibility.

**Algorithm 1** Pseudo-code of L-GMM in a PyTorch-like style.

```
# X: feature embeddings
# K: augmented memory size
# gamma: momentum coefficient
# numGauss: number of Gaussian components for each class
# memory_log: augmented memory for saving log likelihood
# memory_feature: augmented memory for saving feature embeddings

def L-GMM(X, label)
    #== Model Prediction and Training Loss (Eq.16 and Eq.18) ==#

    _c_gauss = MultivariateNormalDiag(means.view(-1, X.shape[1]), scale_diag=
        covariance.view(-1, X.shape[1]))

    probs = _c_gauss.log_prob(X.view(X.shape[0], -1, X.shape[1]))

    unique_c_list = label.unique().long()

    prob_memo_onehot = []
    means_sup = means.data.clone()
    for _c in unique_c_list:

        prob_log_new = probs[label == _c, _c:_c+1, :]
        _c_init_q_log = memory_log[_c:_c+1,:(K - prob_log_new.shape[0]),:]

        # update log_likelihood memory space
        _c_init_q_log = torch.cat([prob_log_new, _c_init_q_log.transpose(0, 1)],
            dim=0)
        _c_init_q_log = _c_init_q_log / _c_init_q_log.sum(dim=-1, keepdim=True)

        # one-hot for best component assignment
        indexs = torch.argmax(_c_init_q_log, dim=-1)
        oneHot_Ver = torch.nn.functional.one_hot(indexs, num_classes=numGauss)
        prob_memo_onehot.append(oneHot_Ver)

        _mem_fea_k = memory_feature[_c:_c+1,:,:].data.clone().transpose(-1,-2)
        n = torch.sum(_c_init_q_log, dim=0)
        n_memo.append(n)

        f = oneHot_Ver.float().permute((1, 2, 0)) @ _mem_fea_k
        f = l2_normalize(f)
        means_sup[_c:_c+1,:self.p_m_n[_c], :] = f

        # encourage the data samples to be evenly distributed
        n_saved = torch.cat(n_memo, dim=1)
        n_supervise = torch.ones_like(n_saved) * (K / numGauss)

    _sum_prob = torch.amax(probs, dim=-1)

    # MLE
    MLE_mask = torch.zeros_like(out_seg)
    for i,j in enumerate(label):
        MLE_mask[i, j] = 1
    MLE = torch.sum(-_sum_prob.mul(MLE_mask))

    losses = CrossEntropyLoss(_sum_prob, label)
    losses -= MLE
    losses['one'] = MSELoss(means, means_sup.float())
    losses['avg'] = WassersteinLoss(n_saved, n_supervise.float())

    return losses
```

## H  DISCUSSION

### H.1  ASSET LICENSE AND CONSENT

We apply three closed-set image classficiation datasets, *i.e.*, CIFAR-10 [43], CIFAR-100 [43] and ImageNet [73], and five open-set image recognition datasets are used, *i.e.*, TinyImageNet (Crop) [47], TinyImageNet (Resize) [47], LSUN (Crop) [100], LSUN (Resize) [100] and iSUN [96].  We use two closed-set semantic segmentation datasets, *i.e.*, ADE20K [109] and Cityscapes [19], and two open-set image segmentation datasets, *i.e.*, Fishyscapes [7] and Road Anomaly [54].  They are all publicly and freely available for academic purposes.  We implement all models with MMClassification [17] and MMSegmentation [18] codebases.  ADE20K (https://groups.csail.mit.edu/vision/datasets/ADE20K/) is released under a CC BSD-3; Cityscapes (https://www.cityscapes-dataset.com/) is released under this License; Road Anomaly (https://www.epfl.ch/labs/cvlab/data/road-anomaly/)) is released under CC BY 4.0; All assets mentioned above release annotations obtained from human experts with agreements.  Fishyscapes (https://fishyscapes.com/) is released under CC BY 4.0, which is synthesized and re-organized from existing datasets that prevents us to trace details; MMClassification (https://github.com/open-mmlab/mmclassification) and MMSegmentation codebases (https://github.com/open-mmlab/mmsegmentation) are released under Apache-2.0.

### H.2  LIMITATIONS AND FUTURE WORK

One limitation of this work is that it currently only considers density-estimating generative models as part of the design. While we believe it is possible to integrate non-density-estimating generative models into this framework, the question remains open for our future endeavors.

We can also naturally extent our work to open-set video segmentation scenarios. Despite progress we have made in current CSR and OSR problems, continuous work should be deployed to delve deeper into the challenges presented by real-time inference [1; 46] and cross-frame/time-step relations [101]. These are aspects often overlooked in image OSR problems, yet they maintain substantial pragmatic relevance in real-world applications. Furthermore, though showing extensive generality across multiple open-set, closed-set datasets, many research [75; 21] have shown that these data have not been closely scrutinized.  For example, ImageNet presents significant geographical and cultural bias, as well as ambiguities [60]. We shall further evaluate our work in dealing with biases learned during training when approaching the open world application.

