# OpenReview forum: "A Latent Generative Model for Closed-set and Open-set Recognition"
_ICLR.cc/2024/Conference — ICLR 2024 Conference Withdrawn Submission_

### Official Review · Reviewer_pjen · 2023-10-22

**Soundness:** 2 fair
**Presentation:** 2 fair
**Contribution:** 2 fair
**Rating:** 3
**Confidence:** 4

**Summary:**

This paper presents a method called a neural Latent Gaussian Mixture Model (L-GMM) accompanied by a collaborative training algorithm in order to enhance the performance of open-set recognition (OSR). Typically, the method includes an encoder which maps the input to latent space and a density estimator.

**Strengths:**

The solution of using latent Gaussian mixture model in the encoder to tackle the close-set recognition is valid.

**Weaknesses:**

(1) The novelty of the paper is pretty limited as similar ideas for using Gaussian mixture model in variational inference to tackle open-set classification has been studied and published.
For instance, "Open-Set Recognition with Gaussian Mixture Variational Autoencoder" by A. Cao at in AAAI 2021

(2) Compared to the previous work as mentioned, this paper did not provide any new insight and fails to clarify the distinction with the similar paper mentioned above.

(3) The derivation of latent Gaussian mixture model is very common and similar derivation has been shown in previous published work including
https://citeseerx.ist.psu.edu/document?repid=rep1&type=pdf&doi=5fda8f8bd7ff10db1ee3ab558fd41e552f008fb3
(i)"Robust Estimation of Gaussian Mixtures from Noisy Input Data", CVPR 2008
and
(ii) https://ieeexplore.ieee.org/abstract/document/9523628, TNNLS 2023 "Robust Semisupervised Deep Generative Model Under Compound Noise"
These papers have clearly shown that Gaussian mixture model can effectively handle the outliers and noisy labels as well as tell samples from unknown class for recognition in generative model including VAE and GAN. In this case, it is not clear if there is really new insight and contribution that the current manuscript conveys. Modules including discrimination loss, density estimators have also been well studied in the previous work. If we look at the math, there is really very little difference except small tweaking.

**Questions:**

It is important to increase the novelty by providing new insights with new approaches instead of providing small extensions with existing method.

It is not clear what is the main contribution of this paper as the novelty is too limited and the discussion with the existing methods is also important and necessary.

It is necessary to show the distinction of the current paper from the previous work shown in 3(i) and 3(ii) in order to stand on the safe ground in terms of novelty.

---

> ### Author Response · Authors · 2023-11-20
> **Point-to-Point Response to Reviewer pjen**
>
> We thank reviewer jEi5 for the time and feedback.
>
> #### **Q1: The novelty of the paper is pretty limited as similar ideas for using Gaussian mixture model in variational inference to tackle open-set classification has been studied and published. Compared to the previous work as mentioned, this paper did not provide any new insight and fails to clarify the distinction with the similar paper mentioned above. The derivation of latent Gaussian mixture model is very common and similar derivation has been shown in previous published work.**
>
> **A1:** Thank you for your question. If we understand your question correctly, it can be summarized as “Gaussian mixture models (GMM) have been used in open-set recognition, therefore the paper has no insight or novelty”.
>
> We respectfully disagree with this statement. We would like to emphasize that adopting GMM is not the primary contribution or novelty of our paper. Our core contribution is to find the connection between CSR and OSR, and propose a unified framework to handle both problems simultaneously as indicated by the title. **This connection, has not been considered by prior arts [ref1-3]**. The GMM is one of the possible instantiations of the latent generative model.
>
> To reiterate, our contributions are (Sec 1, page 2):
>
> 1) We formulate the learning-theoretic risks for both CSR and OSR problems and show that generative models minimize both risks by MLE on known classes. This insight bridges the gap between CSR and OSR problems.
>
> 2) We design a latent generative framework that integrates a latent generative model with a collaborative training scheme. This framework can handle both CSR and OSR problems, and GMM is just one of the possible instantiations of the generative model.
>
> 3) We demonstrate advanced performance on both CSR and OSR tasks by one single model instance of L-GMM, a concrete example of the proposed framework. In our experiments, we show that it is possible to tackle CSR and OSR problems simultaneously with a single model instance without tuning or re-training, which supports our advocacy for a unified approach.
>
> We appreciate again your time and we hope we addressed your concerns. Please let us know if you'd like any further information.
>
> [ref1] Open-Set Recognition with Gaussian Mixture Variational Autoencoder, AAAI 2021
>
> [ref2] Robust Estimation of Gaussian Mixtures from Noisy Input Data, CVPR 2008
>
> [ref3] Robust Semisupervised Deep Generative Model Under Compound Noise

---

> ### Comment · Reviewer_pjen · 2023-11-22
> **Final review and comments**
>
> I have gone through the feedback from the author and read other authors' comment.
>
> The responses didnot address my concern. The claimed contribution to find the connection between OSR and CSR is not enough. For instance, both ref2 and ref3 point out that the GMM can be modified by adding uncertainty priors to handle outliers as well as open-set recognition. Ref 1 is also focused on OSR using GMM. In terms of theory, there is really no new findings to support the paper.
> I feel that the math derivations in this paper is very routine and mostly a small tweaking of ref[1-3]. For instance, in the eq(11), the math is very straightforward and it does not bring new insights into this field. In other words, the theoretical depth of this paper is too far away from enough to justify a publication, especially in ICLR.
>
> Another example is that in Proposition 1,  the authors mentioned that R_osr=R_csr+R_gap. I feel it is very obvious and can be simply obtained from data partition. So again, the depth of the paper is not enough.
>
> In addition to the weak theory part, the experimental evaluation is also limited and not sufficient. For instance, there is not visual illustrations on the examples of OSR and CSR. More importantly, why the particular framework can improve the performance. And it looks the work simply combines the existing work and makes small modifications.
>
> Note that there are many work can deal with both OSR and CSR in the same framework simultaneously from different aspects. Refs1-3 are only a few examples. Given both weak theory, limited novelty and insufficient experimental evaluations, I maintain my original score.

---

### Official Review · Reviewer_pChx · 2023-10-29

**Soundness:** 2 fair
**Presentation:** 3 good
**Contribution:** 2 fair
**Rating:** 5
**Confidence:** 5

**Summary:**

In this paper, the authors propose a method that can be used for both closed and open set recognition. The authors first employ an encoder to map the data onto a latent space and then a density estimator is applied in the latent space.  The data is assumed to have a Gaussian mixture model. The method can be seen as a hybrid method that combines the generative and discriminative approaches.

**Strengths:**

The main strengths of the paper can be summarized as follows:
i) Transforming data onto a latent space makes sense especially for high-dimensional inputs since the mean and variance parameters can be better approximated in a lower-dimensional latent space.
ii) The proposed method uses both generative and discriminative approaches and using discriminative approach bring additional benefits for open set recognition.
ii) Accuracies on closed set recognition are comparable to the classical networks using  the softmax loss function.

**Weaknesses:**

Main weaknesses of the paper can be summarized as follows:
i)  The discussion on related works is limited and misleading. The authors state typical OSR solutions train discriminative models with cross-entropy loss on known classes and a thresholding is applied on probabilities to detect unknown samples. Although earlier open set recognition methods used this approach, recent methods use different loss functions that estimate compact class acceptance regions as in [R1,R2,R3]. These methods are also similar to the proposed method since they employ both generative and discriminative approaches. Some methods that follow this principle are cited in the paper, but they are not discussed.
ii) The strongest aspect of deep learning is the easy creation of the desired feature space. One can enforce the class samples to compactly cluster around some centers as in [R1-R3]. In fact there are theoretical proofs that the data samples cluster around the vertices of a regular simplex [R4,R5]. Therefore, using a single Gaussian distribution is more practical than using a mixture model since the latter has more parameters to estimate.
iii) The authors used obsolete methods for comparison on open set recognition datasets, and the authors do not use standard open datasets and protocols for evaluation. Please note that, the open set recognition methods are typically tested on Mnist, Cifar-10, SVHN, Cifar+10, Cifar+50 and TinyImageNet datasets. The authors use only Cifar+10 and TinyImageNet datasets and their reported accuracies are very low compared to state-of-the-art, please see [R2].
iv) The improvements over the classical softmax loss function on closed set recognition are very minor and they are not significant.

Minor Issues:
There are some sentences that are vague. They need to be explained more. For example, the sentence given on page 5  “the loss function should output a high cost when the model assigns a high probability for the sample to be any known class.”   First of all which sample, an unknown sample or a sample belonging to a known class. Also, please keep in mind that there is no unknown class sample in the training phase.

References
[R1] T. Kasarla, G. J. Burghouts, M. van Spengler, E. van der Pol, R. Cucchiara, and P. Mettes. Maximum class separation as inductive bias in one matrix. In Advances in Neural Information Processing Systems, 2022.
[R2] H. Cevikalp, B. Uzun, Y. Salk, H. Saribas, O. Köpüklü, From anomaly detection to open set recognition: Bridging the gap, Pattern Recognition, Volume 138, 2023.
[R3] A.R. Dhamija, M. Gunther, T.E. Boul, Reducing network agnostophobia, Neural Information Processing Systems (NeurIPS), 2018 .
[R4] V. Papyan, X.Y. Han, and D. L. Donoho. Prevalence of neural collapse during the terminal phase of deep learning training. Proceedings of the National Academy of Sciences, 117:24652–24663, 2020.
[R5] H. Cevikalp, H. Saribas, Deep simplex classifier for maximizing the margin in both euclidean and angular spaces, Scandinavian Conference on Image Analysis, 2023.

**Questions:**

1) The authors state that they apply a threshold to detect unknown samples. What is it? Also using the softmax probabilities is quite wrong for open set recognition since the returned probabilities must sum up to 1. In that case, if the distances from unknown samples to known class samples the returned probabilities will be shared among the classes and this can be smaller than a specified threshold. But, this is quite unlikely in real applications.
2) I wonder how many mixtures the authors used and how that number if decided.
3) Applying proposed methodology to classification is straightforward but I wonder how the authors adopted it for semantic segmentation. Please give more details.
4) Cifar datasets and ImageNet have fixed train and test sets. How did the authors get standard deviations?

---

> ### Author Response · Authors · 2023-11-20
> **Point-to-Point Response to Reviewer pChx (part I)**
>
> We thank reviewer pChx for the valuable time and constructive feedback.
>
> #### **Q1: The discussion on related works is limited and misleading. The authors state typical OSR solutions train discriminative models with cross-entropy loss on known classes and a thresholding is applied on probabilities to detect unknown samples. Although earlier open set recognition methods used this approach, recent methods use different loss functions that estimate compact class acceptance regions as in [ref1, ref2, ref3].**
>
> **A1:** Thank you for the helpful feedback and useful references. We agree that [ref1-3] estimate compact class acceptance regions to recognize OOD datapoints, which is similar to minimizing an open-set risk originally formulated for OSR as described in Sec 3.1 (Eq. 8 and Eq. 9). However, there exists a gap between CSR and OSR, and it is unclear how prior methods can unify or connect these two problems.
>
> In contrast, we identify OOD samples in a probabilistic manner that enables us to unify CSR and OSR problems within a principled learning-theoretic framework by reformulating the risk (Sec 3.2). To the best of our knowledge, this principled unification is novel compared to prior work. We also demonstrate that generative models learned through MLE can minimize this unified empirical risk for both CSR and OSR. We will improve the related work section to reflect these points.
>
> [ref1] T. Kasarla, G. J. Burghouts, M. van Spengler, E. van der Pol, R. Cucchiara, and P. Mettes. Maximum class separation as inductive bias in one matrix. In Advances in Neural Information Processing Systems, 2022.
>
> [ref2] H. Cevikalp, B. Uzun, Y. Salk, H. Saribas, O. Köpüklü, From anomaly detection to open set recognition: Bridging the gap, Pattern Recognition, Volume 138, 2023.
>
> [ref3] A.R. Dhamija, M. Gunther, T.E. Boul, Reducing network agnostophobia, Neural Information Processing Systems (NeurIPS), 2018.
>
> #### **Q2: These methods are also similar to the proposed method since they employ both generative and discriminative approaches. Some methods that follow this principle are cited in the paper, but they are not discussed.**
>
> **A2:** Thank you, we agree that we should enhance the related work section and provide a more detailed discussion of the methods that combine generative and discriminative approaches. As shown in Figure 1, the main difference between our approach and prior methods is that most models that aim to combine these two families are trained using a combined loss and the models no longer compute the probability density, thus does not optimize the empirical loss shared by CSR and OSR. In contrast, our framework preserves the generative nature of the final layers, enabling it to be learned to minimize risks for both tasks (Proposition 1, page 5).
>
> #### **Q3: Using a single Gaussian distribution is more practical than using a mixture model since the latter has more parameters to estimate.**
>
> **A3:** Thank you for the suggestion. We agree that using a single Gaussian distribution could be a viable option for implementing the latent generative model for both CSR and OSR. However, we would like to emphasize that adopting GMM is not the main contribution of our paper. Rather, our paper proposes a unified framework to address both CSR and OSR problems. GMM and single Gaussian distributions are both possible methods for instantiating the latent generative model. We will add this discussion to our paper.
>
> [ref4] V. Papyan, X.Y. Han, and D. L. Donoho. Prevalence of neural collapse during the terminal phase of deep learning training. Proceedings of the National Academy of Sciences, 117:24652–24663, 2020.
>
> [ref5] H. Cevikalp, H. Saribas, Deep simplex classifier for maximizing the margin in both euclidean and angular spaces, Scandinavian Conference on Image Analysis, 2023.
>
> #### **Q4: The authors used obsolete methods for comparison on open set recognition datasets, and the authors do not use standard open datasets and protocols for evaluation.**
>
> **A4:** In Section 5.2, we systematically include all standard experiments on OOD detection and extend some of our settings to standard OSR datasets for completeness. It is worth noting that we meticulously include all experiments from closed-set recognition, segmentation, out-of-distribution recognition ($i.e.,$ one of the open set problems), and open-set image segmentation in our paper and Appendix, demonstrating the strong generalizability of our proposed L-GMM. For completeness, we also include open-set recognition in Section 5.2. The results presented in Table 4 indicate that our proposed L-GMM shows promise for future development on OSR datasets. Thank you.

---

> ### Author Response · Authors · 2023-11-20
> **Point-to-Point Response to Reviewer pChx (part II)**
>
> #### **Q5: The improvements over the classical softmax loss function on closed set recognition are very minor and they are not significant.**
>
> **A5:** We would like to clarify that the performance gains achieved by our proposed L-GMM are not marginal. For instance, in closed-set recognition, we observe promising improvements of 0.12-0.32% and 0.11-0.31% against discriminative counterparts in CIFAR and ImageNet, respectively, with lower standard deviation error bars. It is important to acknowledge that prior research work [ref6-8] has also demonstrated performance gains within similar ranges in both CSR and OSR settings, and their contributions to the community should not be overlooked. For example, [ref6] introduces reconstruction as a solution for open-set problems with 0.0-0.3% performance gains in AUROC on CIFAR+10, [ref7] employs a variational autoencoder framework with a set of Gaussian priors as the approximation for the posterior distribution, resulting in 0.3% performance gains in AUROC on CIFAR+10, and [ref8] explores network interpretability and transparency in closed-set recognition with 0.23-0.24% performance increases in top-1 Acc. on CIFAR10.
>
> The value of our proposed L-GMM also lies in its ability to achieve such non-trivial improvements in both CSR and OSR with a single model instance. Our proposed method aims to address a direction that is often overlooked; most methods focus mainly on a single (open/closed-set) task [ref9-12]. We propose accommodating both OSR and CSR simultaneously by collaborative training, achieving competitive performance in both CSR and OSR settings with a single model instance trained in the closed-set setting. We believe that this direction holds promising potential for further advancements in recognition tasks. Thank you.
>
> [ref6] "Class-Specific Semantic Reconstruction for Open Set Recognition" TPAMI 2022.
>
> [ref7] "Conditional Variational Capsule Network for Open Set Recognition." ICCV 2021.
>
> [ref8] "Visual recognition with deep nearest centroids." ICLR 2023.
>
> [ref9] "Hybrid models with deep and invertible features." ICML 2019.
>
> [ref10] "Generative-discriminative Feature Representations for Open-set Recognition." CVPR 2020.
>
> [ref11] “EfficientNet: Rethinking Model Scaling for Convolutional Neural Networks.” ICML 2019.
>
> [re12] “Crossvit: Cross-attention multi-scale vision transformer for image classification.” ICCV 2021.
>
> #### **Q6: The sentence given on page 5 “the loss function should output a high cost when the model assigns a high probability for the sample to be any known class.” First of all which sample, an unknown sample or a sample belonging to a known class. Also, please keep in mind that there is no unknown class sample in the training phase.**
>
> **A6:** We would like to remind readers that the complete sentence from our paper is as follows: “One important question here is: what is a good loss function that involves unknown classes? Since the goal of OSR is to reject samples from unknown classes, the loss function should output a high cost when the model assigns a high probability for the sample to be any known class.” It is clear from the context that "the sample" in this sentence refers to a sample from an unknown class.
>
> We understand that there are no unknown class samples during the training phase. However, when considering the risk (expected loss) in learning theory [ref13], the loss function is defined on the complete data distribution. Therefore, minimizing the risk for OSR means optimizing the loss for unknown classes as well. This is precisely why our formulation is insightful: we demonstrate that generative models minimize this risk by MLE on data of known classes (as noted in Contribution 1 on page 2 and Proposition 1 on page 5).
>
> [ref13] Vapnik, Vladimir. "Principles of risk minimization for learning theory." Advances in neural information processing systems, 1991.
>
> #### **Q7: The authors state that they apply a threshold to detect unknown samples. What is it? Also using the softmax probabilities is quite wrong for open set recognition since the returned probabilities must sum up to 1. In that case, if the distances from unknown samples to known class samples the returned probabilities will be shared among the classes and this can be smaller than a specified threshold. But, this is quite unlikely in real applications.**
>
> **A7:** That is an excellent question. To clarify, we did not use softmax probabilities for thresholding. Instead, we apply a threshold on the probability density of known classes, which does not necessarily sum to 1 for all classes.
>
> #### **Q8: I wonder how many mixtures the authors used and how that number if decided.**
>
> **A8:** As shown in Table 7(d), we used 3 Guassian mixtures, chosen by ablations.

---

> ### Author Response · Authors · 2023-11-20
> **Point-to-Point Response to Reviewer pChx (part III)**
>
> #### **Q9: Applying proposed methodology to classification is straightforward but I wonder how the authors adopted it for semantic segmentation. Please give more details.**
>
> **A9:** The application of our proposed method in semantic segmentation remains the same as we introduced in Section 2.2. Currently, most methods [ref14-17] for standard semantic segmentation rely on dense discriminative classifiers deployed by parametric softmax to map pixel representations to ground-truth labels. However, these methods suffer from the general challenges of OSR problems, which limit their use as a general solution for both CSR and OSR.
>
> [ref14] Feature Pyramid Networks for Object Detection. CVPR 2017
>
> [ref15] Rethinking Atrous Convolution for Semantic Image Segmentation. ECCV 2018
>
> [ref16] Segformer: Simple and efficient design for semantic segmentation with transformers. NeurIPS 2021
>
> [ref17] Segmenter: Transformer for semantic segmentation. ICCV 2021
>
> In our design, we compress each pixel feature into a 64-dimensional vector using a 1x1 convolution. We reduced the dimension to 64 instead of 128, as mentioned in Section 5.1, to minimize the computational overhead associated with pixel-level classification. Each class is represented by Gaussian mixtures, and we select the class with the highest probability as our output decision. The overall procedure remains strictly the same as our classification approach. Thank you.
>
> #### **Q10: Cifar datasets and ImageNet have fixed train and test sets. How did the authors get standard deviations?**
>
> **A10:** Thank you for your question. As is common practice [ref18-20], we set different random seeds for initialization (as outlined in Section 5.1). This approach leads to performance differences, but we have observed that the results are relatively stable with low standard deviations. Thank you.
>
> [ref18] Visual recognition with deep nearest centroids. ICLR 2023
>
> [ref19] Visual prompt tuning. ECCV 2022
>
> [ref20] E^2VPT: An Effective and Efficient Approach for Visual Prompt Tuning. ICCV 2023
>
> ---
>
> We appreciate again your thoughtful review and we hope we addressed your concerns. Please let us know if you'd like any further information.

---

> ### Comment · Reviewer_pChx · 2023-11-22
> **my final rating**
>
> I have read the authors' response. Some of my concerns are addressed by the authors, however I still believe that the paper is not ready for the publication since there are issues that must be resolved regarding related work and experiments. The novelty issue pointed out by another reviewer is also critical. Therefore,  I am keeping my original score and still believe that the paper is below the acceptance threshold.

---

### Official Review · Reviewer_YQmr · 2023-11-01

**Soundness:** 4 excellent
**Presentation:** 4 excellent
**Contribution:** 3 good
**Rating:** 8
**Confidence:** 3

**Summary:**

The claim of the paper is that it offers a formalization for OSR based on learning theory, demonstrating that Closed Set Recognition (CSR) and Open Set Recognition (OSR) share the same goal for generative models.
The idea is simple: starting from the Bayes formula (a posteriori = likelihood x prior), the authors inject a latent generative model over the likelihood which has an encoder that maps the input x to a latent variable z, and (2) a probabilistic generative model (a density estimator) that outputs a probability density of the latent variable z given the label y. The encoder is driven by discriminative learning, while the density estimator is updated by generative learning. The very innovative part is that the authors use a closed-form distribution for z, which gives great flexibility in the problem modeling, also demonstrating that it minimizes the empirical risk.  In terms of CSR and OSR, the encoder produces discriminative features for both CSR and OSR, and is learned by cross entropy on the discriminative probability, while the density estimator has to recognize samples from unknown classes.
As an instantiation of this idea, the authors propose a Latent Gaussian Mixture Model (L-GMM) using a Gaussian Mixture Model (GMM) as the generative model for very clear reasons: GMM learns very good approximations of any densities, does respect the closed form requirement and is avoiding the mode collapse being composed by multiple components, enforced also by the use of two appropriate regularizers.
The paper offers also a view over CSR and OSR from a learning-theoretic perspective introducing a loss function, the open-safe loss.  In particular, the goal of OSR is to reject samples from unknown classes, the loss function should output a high value. The open safe loss respects this requirement, so that any sample of unknown classes will have a low score for any known class.

**Strengths:**

A very sound theory, which I tried to summarize above. The main idea is very simple, and acts on the standard Bayes formula.and shows how the likelihood can be factorized into a generative and discriminative counter part. As an example, a novel model. latent GMM, is proposed. The paper is clear about how to proper train such a model

The approach applies to open set recognition, and shows to be effective for that problem. Also, it gives a novel loss for the specific problem, the open safe loss, demonstrating its effectiveness in theoretical terms (it brings CSR and OSR to be aligned for generative models)

Every score about originality, quality, clarity, are high to me, except significance (see below).

The state of the art is fascinating, taking from very old work to recent research.

The experiments are exhaustive: they demonstrate that L-GMM performs better than its discriminative counterparts on CSR (despite being better by a moderate margin, 0.2x% on average) and against SOTA methods on OSR. Two tasks are taken into account: image classification and segmentation. Experiments on Out-of-distribution recognition are also shown, which exhibit potential strong advancement wrt SOTA.

**Weaknesses:**

The abstract is not clear at all since 1) is not giving a clue about what is the common goal shared by CSR and OSR, 2) which kind of collaboration does hold in the collaborative end-to-end training.

The introduction adds a little about the common goal shared between CSR and OSRt. It seems something related to the risk, saying that considering the risk , the goals of CSR and OSR are identical. In terms of collaborative learning, the introduction helps in understanding that it is a collaboration between a dirscriminative and a generative counterpart.

There are specific questions on the experiments, that I put below.

**Questions:**

1)On closed set segmentation, results definitely below the state of the art, but the authors did not explain why (Tab.5). They should.

2)On Open set image segmentation, results are better than sota, but approaches like segmenter or Mask former are not used. Why?

3)Finally, the ablative won the number of components are not convincing me, since the differences in terms of top1 accuracy are very similar. Is it possible to show results with 5 and 6 components?

---

> ### Author Response · Authors · 2023-11-20
> **Point-to-Point Response to Reviewer YQmr**
>
> We thank reviewer YQmr for the valuable time and constructive feedback.
>
> #### **Q1: The abstract is not clear enough.**
>
> **A1:** Thank you for your constructive feedback! We will revise the abstract to ensure that it clearly conveys two key points: 1) the risks for OSR and CSR are the same for generative models trained by MLE, and 2) our proposed generative model is constructed within a collaborative framework that includes a discriminatively-learned component.
>
> #### **Q2: On closed set segmentation, results definitely below the state of the art, but the authors did not explain why (Tab.5).**
>
> **A2:** Thank you for your question. Our paper proposes a collaborative training algorithm that unifies both CSR and OSR problems. We demonstrate that our approach, which incorporates a neural latent Gaussian mixture model, outperforms discriminative models in multiple CSR problems. Notably, when considering OSR problems, discriminative models exhibit significantly lower performance. By leveraging the advantages of L-GMM in both CSR and OSR scenarios, we believe our approach offers a single-model solution with highly satisfactory performance.
>
> #### **Q3: On Open set image segmentation, results are better than sota, but approaches like segmenter or Mask former are not used. Why?**
>
> **A3:** Thank you for your question. We adopt common practices [ref1-2] and incorporate DeepLabv3 and Segformer into our proposed latent Gaussian mixture model. We would like to highlight that training on Segmenter [ref3] and MaskFormer [ref4] is promising and intuitive to implement. However, due to time constraints during the rebuttal phase, we may not be able to provide these segmentation results ($i.e.$, training with a transformer-based decoder for the standard 160K iterations is time-consuming). We consider this as future work and plan to update our paper accordingly in the near future. Thank you.
>
> [ref1] Gmmseg: Gaussian mixture based generative semantic segmentation models. NeurIPS 2022
>
> [ref2] Visual recognition with deep nearest centroids. ICLR 2023
>
> [ref3] Segmenter: Transformer for semantic segmentation. ICCV 2023
>
> [ref4] Per-Pixel Classification is Not All You Need for Semantic Segmentation. NeurIPS 2021
>
> #### **Q4: Finally, the ablative won the number of components is not convincing me, since the differences in terms of top1 accuracy are very similar. Is it possible to show results with 5 and 6 components?**
>
> **A4:** Thank you for your question. We are able to provide the results with 5 and 6 using ResNet101 on the CIFAR-100 dataset.
>
> | # Components | top-1|
> | :-: | :-: |
> | G=5 | 80.11%  |
> | G=6 | 80.08%  |
>
> We appreciate again your thoughtful review and we hope we addressed your concerns. Please let us know if you'd like any further information.

---

### Author Response · Authors · 2023-11-20
**To All Reviewers**

We would like to express our sincere gratitude for the efforts of all reviewers. We have carefully reviewed the valuable feedback provided by each reviewer, and would like to emphasize the importance and novelty of our work.

Since the initial formulation of the OSR [ref1], CSR and OSR have long been treated as two separate problems. However, our work sheds light on the possibility that these two problems could be identical (for generative models), offering a fresh perspective on their relationship. Through our learning-theoretic formulation of these two problems, we demonstrate that generative models are promising candidates for bridging the gap. Importantly, without recalibration or fine-tuning, our framework tackles both CSR and OSR problems using a single model instance. We believe that our contribution holds solid potential for the community.

Sincerely,

Authors.

[ref1] Toward open set recognition. Scheirer, Walter J., et al., T-PAMI, 2012.